# The Prognostic Value of the De Ritis Ratio for Progression-Free Survival in Patients with NET Undergoing [^177^Lu]Lu-DOTATOC-PRRT: A Retrospective Analysis

**DOI:** 10.3390/cancers13040635

**Published:** 2021-02-05

**Authors:** Tristan Ruhwedel, Julian M. M. Rogasch, Kai Huang, Henning Jann, Imke Schatka, Christian Furth, Holger Amthauer, Christoph Wetz

**Affiliations:** 1Department of Nuclear Medicine, Charité—Universitätsmedizin Berlin, Corporate Member of Freie Universität Berlin, Humboldt—Universität zu Berlin, and Berlin Institute of Health, Augustenburger Platz 1, 13353 Berlin, Germany; tristan.ruhwedel@charite.de (T.R.); julian.rogasch@charite.de (J.M.M.R.); kai.huang@charite.de (K.H.); imke.schatka@charite.de (I.S.); christian.furth@charite.de (C.F.); holger.amthauer@charite.de (H.A.); 2Berlin Institute of Health (BIH), 13353 Berlin, Germany; 3Department of Hepatology and Gastroenterology, Charité—Universitätsmedizin Berlin, Corporate Member of Freie Universität Berlin, Humboldt—Universität zu Berlin, and Berlin Institute of Health, Augustenburger Platz 1, 13353 Berlin, Germany; henning.jann@charite.de

**Keywords:** De Ritis ratio, NET, neuroendocrine tumor, CgA, Chromogranin A, AST, ALT, PRRT, peptide receptor radio nuclide therapy, DOTATOC

## Abstract

**Simple Summary:**

Peptide receptor radionuclide therapy (PRRT) of neuroendocrine tumors (NET) has shown variable response rates between 9% and 39%. Therefore, better criteria are needed that help doctors to identify patients who will show a favorable outcome to PRRT, and which patients may not. The so-called De Ritis ratio, which is calculated using two basic laboratory parameters of liver function, has shown that it can help to predict the patient outcome in various tumor types. This retrospective study included 125 patients with NET who were treated with PRRT. We demonstrated that a high De Ritis ratio and high levels of the tumor marker Chromogranin A (CgA) each improved the prediction of the progression-free survival after treatment. A consequence for clinical care might be that patients with both high De Ritis ratio and high CgA levels may benefit from intensified follow-up imaging after PRRT because they have a higher risk of early progression.

**Abstract:**

Background: The De Ritis ratio (aspartate aminotransferase [AST]/alanine aminotransferase [ALT]) has demonstrated prognostic value in various cancer entities. We evaluated the prognostic capability of the De Ritis ratio in patients with metastatic neuroendocrine tumors (NET) undergoing peptide receptor radionuclide therapy (PRRT). Methods: Unicentric, retrospective analysis of 125 patients with NET undergoing PRRT with [^177^Lu]Lu-DOTATOC (female: 37%; median age: 66 years; G1+G2 NET: 95%). The prognostic value regarding progression-free survival (PFS) was analyzed with univariable and multivariable Cox regression. Prognostic accuracy was determined with Harrell’s C index and a likelihood ratio test. Results: Progression, relapse, or death after PRRT was observed in 102/125 patients. Median progression-free survival (PFS) was 15.8 months. Pancreatic or pulmonary origin, high De Ritis ratio, and high Chromogranin A (CgA) significantly predicted shorter PFS in univariable Cox. In multivariable Cox regression, only high De Ritis ratio >0.927 (HR: 1.7; *p* = 0.047) and high CgA >twice the upper normal limit (HR: 2.1; *p* = 0.005) remained independent predictors of shorter PFS. Adding the De Ritis ratio to the multivariable Cox model (age, Eastern Cooperative Oncology Group (ECOG) performance status, primary origin, CgA) significantly improved prognostic accuracy (*p* < 0.001). Conclusions: The De Ritis ratio is simple to obtain in clinical routine and can provide independent prognostic value for PFS in patients with NET undergoing PRRT.

## 1. Introduction

Neuroendocrine tumors (NET) of the bronchopulmonary (lung-) or gastroenteropancreatic system (GEP-NET) represent a rare and heterogeneous class of tumors [1]. During the last few decades, the incidence of NET has significantly increased more than six-fold, from 1.1/100,000 persons to 7.0/100,000 persons, mainly due to the progress in functional imaging, device-specific sensitivity, and further increasing awareness of the occurrence of NET [1,2].

For metastatic well-differentiated GEP-NET of low and intermediate grade G1 and G2, peptide receptor radio nuclide therapy (PRRT) with [^177^Lu]Lu-[DOTA^0^-Tyr^3^]octreotate ([^177^Lu]DOTATOC) is a well-established second- to third-line therapy after progress under treatment with common “cold” somatostatin analogues [3]. The NETTER-1 study, the first randomized controlled trial in patients with metastasized GEP-NET and treatment with PRRT, showed an average therapy response rate of 18% in these patients [4]. However, prognosis varies, and not all patients benefit from PRRT. Many retrospective studies found response rates between 9 and 39% [5,6]. A recent meta-analysis including 11 studies and 1268 patients reported an average response rate of 29.1% [7]. Therefore, better stratification criteria are highly desirable to identify patients who will ultimately show a favorable response and longer progression-free survival (PFS) after PRRT.

As of today, tissue measurements of the well-established proliferation-marker Ki-67 and serum levels of the neuroendocrine secretory protein Chromogranin A (CgA) are the best prognostic indicators for NET patients undergoing PRRT [8,9,10,11]. Recently, Aalbersberg et al. demonstrated that higher CgA levels as well as higher Ki-67 values were statistically significantly associated with shorter PFS [10]. Corresponding observations have also been made by other authors [8,9,12,13]. In addition, biological somatostatin receptor (SSR) diversities (RADIOMICS) [14], especially the spatial nonuniformity of the functional lesion volume in SSR-imaging have recently been investigated for their prognostic value [15,16,17,18,19].

The ratio of aspartate aminotransferase [AST]/alanine aminotransferase [ALT] in the pretherapeutic blood serum or heparin plasma, the so-called ‘’De Ritis ratio’’ [20], has been recently reported to be a valuable and independent prognostic factor in the treatment of different tumor entities [21,22,23,24,25,26,27,28,29]. Furthermore, a representative meta-analysis, which included 9400 patients with different types of cancer, demonstrated that a high AST/ALT ratio was associated with an impaired overall survival (OS) [30]. Until now, the De Ritis ratio has not been addressed in the context of NETs. The primary aim of this study was to evaluate the prognostic value of the pretherapeutic De Ritis ratio on the PFS in patients with NET undergoing PRRT. Secondly, a combined risk score of high De Ritis ratio and high CgA was evaluated to predict the PFS.

## 2. Results

### 2.1. Patients

In 64/125 patients (51%), the primary tumor was located in the gastrointestinal tract, 30/125 patients (24%) showed a pancreatic primary, 11/125 patients (9%) had a pulmonary primary, and 20/125 patients (16%) suffered from cancer of unknown primary (CUP). Table 1 illustrates all patient characteristics.

### 2.2. Progression Free Survival

During follow-up, disease progression or relapse was observed in 102 patients (82%), and median PFS for the total cohort was 15.8 months (interquartile range [IQR]: 8.2–28.5 months; Figure 1). The median follow-up duration in patients without disease progression/relapse was 19 months (IQR: 15.8–25.0 months). No nephrotoxicity grade ≥3, hematologic toxicity grade ≥3, tumor lysis syndrome, or dose-limiting liver damage were observed.

Patients with a high De Ritis ratio (>0.927) had a significantly shorter PFS (median: 14.7 months; IQR: 7.1–24.3 months) than patients with low De Ritis ratio (median: 24.3 months; IQR: 8.5–38.4 months; log-rank test: *p* = 0.006; Figure 2a). Patients with a high CgA (>204 µg/L) also had significantly shorter PFS (median: 13.1 months; IQR: 6.7–22.5 months) than those with low CgA (median: 26.9 months; IQR: 16.6–38.4 months; *p* = 0.001; Figure 2b).

In univariable Cox regression (Table 2), CgA >204 μg/L (hazard ratio [HR]: 2.16; 95% confidence interval [95% CI]: 1.35–3.46; *p* = 0.001) and De Ritis ratio >0.927 (HR: 1.89; 95% CI: 1.19–3.0; *p* = 0.007) were significant predictors for shorter PFS. Compared to patients with gastrointestinal primaries, patients with pancreatic (HR: 1.8; 95% CI: 1.11–2.94; *p* = 0.018) or pulmonary primaries (HR: 2.62; 95% CI: 1.30–5.28; *p* = 0.007) had significantly impaired PFS. Age >66 years (HR: 1.46; 95% CI: 0.98–2.18; *p* = 0.061) and an Eastern Cooperative Oncology Group (ECOG) score of 2 (HR: 4.0; 95% CI: 0.95–16.84; *p* = 0.059) showed a trend towards an association with shorter PFS but did not reach statistical significance.

In multivariable Cox regression (Table 2), only De Ritis ratio >0.927 (HR: 1.7; 95% CI: 1.01–2.86; *p* = 0.047) and CgA >204 μg/L (HR: 2.05; 95% CI: 1.24–3.39; *p* = 0.005) remained as independent predictors of shorter PFS.

### 2.3. Predictive Model for Progression-Free Survival

The Cox model, including high age, ECOG score, primary tumor location, and high CgA (model 1), showed a likelihood ratio (LR) χ^2^ of 11.5 and Harrell’s C of 0.633. The predictive accuracy was slightly but significantly higher if the Cox model included high age, ECOG score, primary tumor location and the combined score of high CgA and high De Ritis ratio (model 2; LR χ^2^ = 14.0, Harrell’s C = 0.65, LR test: *p* < 0.001).

Combining the two factors high CgA and high De Ritis ratio in a prognostic score, the 70 patients (56%) with both factors showed a median PFS of 12.4 months (IQR: 6.6–22.1 months) compared to 42 patients (34%) with one of both factors (median PFS: 20.4 months; IQR: 8.5–33.8 months) and 13 patients (10%) with none of either factor (median PFS: 34.0 months; IQR: 24.3–38.4 months; log-rank test: *p* < 0.001; Figure 3).

## 3. Discussion

The aim of this study was to assess the prognostic value of the pretherapeutic De Ritis ratio regarding PFS in patients with NET undergoing treatment with [^177^Lu]Lu-DOTATOC PRRT. To the best of our knowledge, there has not been a previous study analyzing the prognostic value of the De Ritis ratio in the context of NET. However, Wu et al. included 8853 patients with various tumor entities in a meta-analysis and illustrated the additional value of the De Ritis ratio as a prognostic parameter for survival outcomes (OS: HR = 1.7, *p* < 0.001) [30].

At present, a compelling explanation of the prognostic value of the De Ritis ratio remains to be found [20]. Both enzymes AST and ALT are routinely determined as parts of the “liver function panel” [31]. The physiologic AST/ALT ratio in hepatocytes is 2.5/1, while ALT presents a two-fold increased biological half-life (t½ = 36 h) in comparison to AST (t½ = 18 h) [31]. Therefore, the concentration of these enzymes should be equalized in the blood over the long term [31]. However, in the case of an increased rate of hepatocyte apoptosis, the serum ratio of AST/ALT (De Ritis ratio) rises as more AST and ALT are released than eliminated according to the half-life [31,32]. Nonetheless, it should be noted that AST is expressed in various tissue types, while ALT is more liver-specific [31]. Consequently, an increased AST is not always caused by hepatic pathology, especially if ALT is not comparably and simultaneously elevated [31,33]. It is currently hypothesized that anaerobic glycolysis, which is typical of the metabolism of cancer cells (the “Warburg effect”) may explain the prognostic value of the De Ritis ratio [21,22,23,24,26,27,28,29,30,34,35]. In this context, several interactions exist between increased anaerobic glycolysis, an altered NAD+/NADH ratio in the cytoplasm, and AST, which is essential for the function of the malate-aspartate shuttle [31,36,37,38]. Underlining this hypothesis, Thornburg et al. demonstrated that cancer cells are especially dependent on AST for a high proliferation rate [39]. In the current sample, we therefore suspect that the elevated De Ritis ratio in the corresponding subgroup was more likely related to a higher tumor proliferation rate than to a hepatic source, as ALT was significantly lower in this subgroup. 

In our analysis, a high pretherapeutic De Ritis ratio was a significant predictor of shorter PFS in multivariable Cox regression. Furthermore, the Harrell’s C index showed that the combined model with a high De Ritis ratio outperformed the prognostic accuracy of clinical factors and high CgA alone. Recently, the De Ritis ratio was introduced as a prognostic parameter of PFS, OS, and other survival outcomes in various tumor types [21,22,23,24,25,26,27,28,29,30]. In particular, Bezan et al. firstly attempted a differentiation in low and high De Ritis ratio and demonstrated the prognostic capability of the De Ritis ratio in patients with localized renal cell carcinoma (OS: De Ritis ≥1.26, HR = 1.76, *p* < 0.001) [21]. Moreover, Wang et al. created a predictive model combining the De Ritis ratio with the Gleason score and pathological tumor stage in patients with localized prostate cancer and showed that this model predicted biochemical recurrence-free survival [22]. In most previous studies, the cut-off for a high De Ritis ratio was usually 1.1 to 1.65, which was higher than the optimal cut-off in the current analysis (>0.927) [21,22,23,25,26,27,28,29]. Comparing the cut-off levels, differences could be caused by the different tumor entities, clinical settings and treatment strategies as well as different patterns of metastases, especially with respect to the hepatic metastasis burden [21,22,23,24,25,26,27,28,29,30]. Furthermore, prospective validation of cut-offs was generally not reported, potentially limiting their comparability [21,22,23,24,25,26,27,28,29,30]. 

Additionally, we could report that a high pretherapeutic CgA was a significant predictor of shorter PFS in multivariable Cox regression. Previously, several authors could demonstrate the prognostic value of CgA in patients with NET undergoing treatment with PRRT [10,12,13]. In a recent analysis by Aalbersberg et al., patients with a CgA ≥336 μg/L (median) had a significantly shorter PFS in multivariable Cox regression than patients with a CgA <112 μg/L (first quartile) [10]. These studies are well in line with our finding, which found that patients demonstrated a significantly shorter PFS with CgA values > twice the upper normal limit (2xULN). The median of our analysis, nevertheless, was almost two times higher (336 vs. 612 μg/L). However, we used a different cut-off (2xULN) as this cut-off was already validated in a prospective, multinational, phase 2 study [40]. Differences may arise, since Aalbersberg et al. only included patients who received at least 3 cycles of PRRT [10]. CgA values should be generally compared cautiously since they may differ between laboratories [41].

This study was limited by its retrospective nature and lack of a matching control group of patients undergoing a different treatment. Therefore, a predictive capability of the De Ritis ratio could not be formally assessed. Prospective studies are required to validate the current explorative results and to ensure a well-selected, homogenous patient collective. Moreover, due to the retrospective setting, we could not assess descriptive follow-up parameters of laboratory values or imaging findings beyond clinical routine data, which should be addressed in subsequent prospective studies. As recently demonstrated, values for AST, ALT, and CgA might increase after administration of PRRT cycles [42,43]. Given the retrospective nature of this study design, data on the short-term dynamic of these laboratory values were not available to detect these reversible alterations. Therefore, prospective evaluation with close post-therapeutic monitoring of the parameters may be useful to investigate the prognostic relevance of their post-therapeutic dynamic. Notably, this would be independent from the current observation of their prognostic value prior to treatment. Furthermore, patients with a high De Ritis ratio were, on average, significantly older and showed significantly higher CgA values (Table 1). In principle, this might introduce a bias in evaluating a prognostic value of the De Ritis ratio. However, multivariable Cox regression demonstrated that an independent prognostic value of the De Ritis ratio remained after adjustment for these other risk factors. A multivariable Cox regression in an unmatched cohort was favored over a matched-pair analysis, e.g., using propensity score matching as conducted by other groups [27,29]. As reported by Biondi-Zoccai et al., multivariable Cox regression should be preferred over propensity score matching if the ratio of events to variables is >8–10 (as in the current analysis), because matching usually involves discarding patients without a proper match. This reduces statistical power [44], which other researchers have confirmed [45].

Determination of both CgA and De Ritis ratio is inexpensive and already part of a clinical routine in patients with NET. Therefore, the necessary validation of the current results in prospective (multicenter) studies should be straightforward. Moreover, proof of a predictive value of De Ritis ratio, in addition to its prognostic significance [46], will be required, as only this will have direct impact on clinical decisions among a broad spectrum of therapeutic strategies for patients with well-differentiated NET.

## 4. Materials and Methods 

### 4.1. Patient Population

This retrospective, unicentric study analyzed 125 patients with histologically proven NET treated with PPRT between September 2007 and October 2019. All patients fulfilled the following inclusion criteria: (1) Progressive, metastasized NET, (2) positivity for SSR expression in functional imaging (SSR-positron emission tomography/computed tomography (PET/CT) or SSR scintigraphy), (3) De Ritis ratio determined immediately before application of the first cycle of PRRT, (4) a follow-up after application of the first cycle PRRT ≥12 months and (5) no myocardial infarction ≤14 days before application of the first cycle PRRT [33]. In 121 of 125 patients (97%), PRRT followed previous treatments (operative resection: n = 78; somatostatin analogues: n = 87; mTOR inhibitor: n = 24; tyrosine kinase inhibitor: n = 7; chemotherapy: n = 38; local ablative therapy: n = 10; radiation therapy: n = 6; transcatheter arterial chemoembolization: n = 9).

### 4.2. [^177^Lu]Lu-DOTATOC-PRRT and Response Assessment

Patients underwent PRRT with a median of 3 cycles (range: 1–6 cycles) and a scheduled dose of 200 mCi (7.45 GBq) [^177^Lu]Lu-DOTATOC per cycle. PRRT cycles were administered in intervals of 10 to 12 weeks. After application of two cycles of PRRT all patients underwent SSR-PET/CT with [^68^Ga]Ga-DOTATOC as interim staging for response evaluation, which was repeated every two cycles. In addition, interim staging was generally performed at least 2 months after application of the last cycle PRRT to avoid misinterpretation due to possible pseudo-progression (radiogenic edema) [42]. Progressive disease was determined by an interdisciplinary tumor board. In the case of progressive disease, no further PRRT cycles were administered. After treatment completion, patients underwent follow-up imaging every 3 to 6 months. Morphological assessment was generally performed by contrast-enhanced (CE) CT. Alternatively, CE magnetic resonance imaging (MRI) was used for morphological evaluation if available.

### 4.3. Evaluation

AST, ALT, and CgA were determined <4 weeks before application of the first cycle of PRRT (De Ritis ratio = ALT/AST). PFS was defined as the time from the first cycle of PRRT until detection of progressive disease according to the response evaluation criteria in solid tumors (RECIST) 1.1 or death from any cause [47].

### 4.4. Statistical Analysis

Statistical analysis was performed using SPSS version 25 (IBM, Chicago, IL, USA) and R 4.0.0 (Foundation for Statistical Computing, Vienna, Austria, 2020; http://www.R-project.org (accessed on 19 January 2020)). Significance was assumed at ɑ = 0.05. Descriptive values were expressed as median, IQR, and range. Univariable Cox proportional hazards regression regarding the PFS included clinical parameters (sex, age, functionality of the NET, presence of a Hedinger syndrome, localization of the primary tumor (gastrointestinal, pancreatic, pulmonary, cancer of unknown primary), localization of metastases (hepatic, lymphonodal, osseous, peritoneal, pulmonary), ECOG score [48] and Charlson comorbidity index (CCI) [49]) and laboratory parameters (CgA, De Ritis ratio).The HR and the 95% CI were determined for each parameter. Before inclusion, continuous variables (age, CgA, De Ritis ratio) were binarized. The cut-off for age (>66 years) was defined as the median value in the patient sample, while the CgA cut-off (>204 µg/L) was 2xULN [40,50]. The De Ritis ratio was binarized with a cut-off (>0.927) that achieved the minimum *p*-value in the log-rank test as determined with the Charité Cutoff Finder [51]. Variables were compared between groups with low vs. high De Ritis ratio using the Wilcoxon rank-sum test (continuous variables) or Fisher’s exact test (categorical variables). All variables with *p* ≤ 0.1 in univariable Cox regression were also candidates for inclusion into multivariable Cox regression. The proportional hazard assumption was tested using the goodness-of-fit test and fulfilled by each variable. Using equal weights, the factors high CgA and high De Ritis ratio were combined as a predictive model regarding PFS. Using the rms package for R, the LR χ^2^ and Harrell’s C index of the multivariable Cox model were calculated after inclusion of either high age, ECOG, primary tumor location, and high CgA (model 1), or high age, ECOG, primary tumor location, and the combined score of high CgA and high De Ritis ratio (model 2) [52]. The likelihood ratio test for these two Cox models was performed to test if the combined score of CgA and De Ritis ratio provides additional prognostic value over CgA alone. The Kaplan–Meier method was used to estimate survival rates and average PFS.

## 5. Conclusions

The De Ritis ratio provided an independent prognostic value for PFS in patients with NET undergoing PRRT with [^177^Lu]Lu-DOTATOC. Consequently, the follow-up in patients with both a high De Ritis ratio and high CgA might be intensified as they have a higher risk for early progression. Its assessment as a routine clinical laboratory parameter is straightforward, while the cause of its prognostic value remains unclear, but likely relates to tumor metabolic activity instead of liver function, per se.

## Figures and Tables

**Figure 1 cancers-13-00635-f001:**
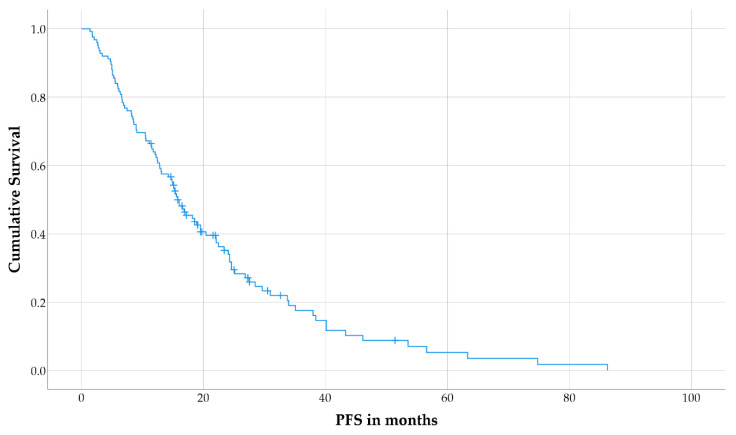
Kaplan–Meier curve for progression-free survival (PFS) in the total cohort.

**Figure 2 cancers-13-00635-f002:**
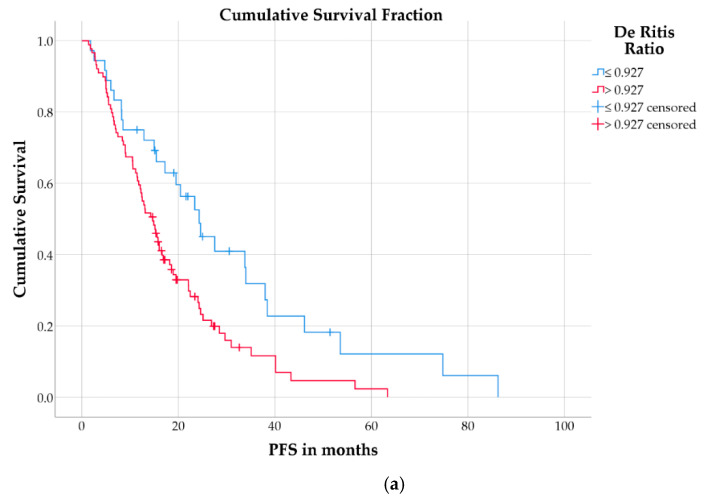
Kaplan–Meier curves for PFS in patients separated by (**a**) De Ritis ratio or (**b**) CgA.

**Figure 3 cancers-13-00635-f003:**
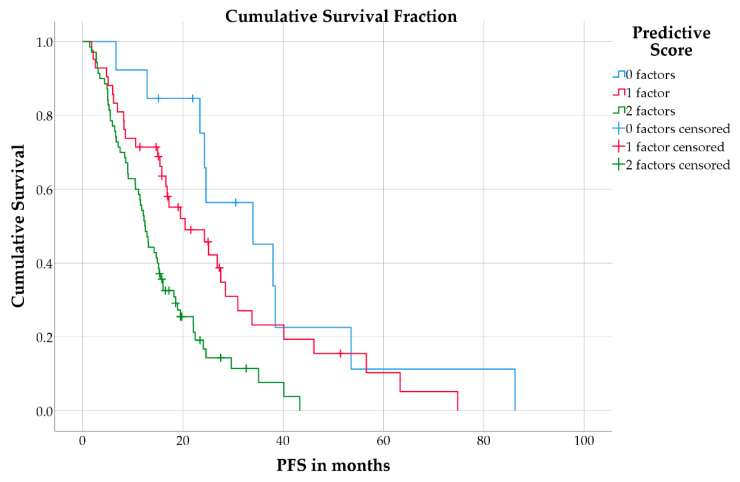
Kaplan–Meier curves for PFS in patients separated by the combined risk score.

**Table 1 cancers-13-00635-t001:** Patient characteristics.

Variable	*n* (%) or Median (Range)			*p*
	Total Cohort	De Ritis Ratio Low	De Ritis Ratio High	
Patient count	125	36	89	
Age in years	66 (35–87)	59 (37–80)	69 (35–87)	<0.001
Age >66 years	65 (52%)	8 (22%)	57 (64%)	<0.001
Age ≤66 years	60 (48%)	28 (78%)	32 (36%)	
Sex				0.42
Men	79 (63%)	25 (69%)	54 (61%)	
Women	46 (37%)	11 (31%)	35 (39%)	
ECOG score				0.68
0	90 (72%)	28 (78%)	62 (70%)	
1	33 (26%)	8 (22%)	25 (28%)	
2	2 (2%)	-	2 (2%)	
Charlson comorbidity index (CCI)	1 (0–7)	1 (0–7)	1 (0–6)	0.94
Primary location				0.57
Gastrointestinal	64 (51%)	22 (61%)	42 (47%)	
Pancreatic	30 (24%)	8 (22%)	22 (25%)	
Pulmonal	11 (9%)	2 (6%)	9 (10%)	
CUP	20 (16%)	4 (11%)	16 (18%)	
Metastatic disease	124 (99%)	36 (100%)	88 (99%)	
Metastatic spread				
Hepatic	119 (95%)	32 (89%)	87 (98%)	0.057
Lymphonodal	104 (83%)	25 (69%)	79 (89%)	0.016
Osseous	51 (41%)	12 (33%)	39 (44%)	0.32
Peritoneal	24 (19%)	4 (11%)	20 (23%)	0.21
Pulmonal	6 (5%)	2 (6%)	4 (5%)	1.0
Functional tumor	42 (34%)	11 (31%)	31 (35%)	0.68
Hedinger syndrome	5 (4%)	1 (3%)	4 (5%)	1.0
Grading				0.65
G1	24 (19%)	5 (14%)	19 (21%)	
G2	95 (76%)	29 (81%)	66 (74%)	
G3	6 (5%)	2 (6%)	4 (5%)	
Laboratory parameters				
Chromogranin A in μg/L	612 (14–601,700)	355 (14–5283)	751 (38–601,700)	0.027
Chromogranin A >204 μg/L	93 (74%)	23 (64%)	70 (79%)	0.11
Chromogranin A ≤204 μg/L	32 (26%)	13 (36%)	19 (21%)	
AST in U/L	29 (13–139)	30 (20–84)	28 (13–139)	0.34
ALT in U/L	28 (10–132)	46 (23–122)	23 (10–132)	<0.001
De Ritis ratio	1.09 (0.46–2.87)	0.74 (0.46–0.93)	1.23 (0.93–2.87)	
Number of PRRT cycles	3 (1–6)	3 (1–5)	3 (1–6)	0.084
Previous treatment				
Operative resection	78 (62%)	22 (61%)	56 (63%)	0.84
Somatostatin analogues	87 (70%)	20 (56%)	67 (75%)	0.034
mTOR inhibitor	24 (19%)	9 (25%)	15 (17%)	0.32
Tyrosine kinase inhibitor	7 (6%)	3 (8%)	4 (5%)	0.41
Chemotherapy	38 (30%)	10 (28%)	28 (32%)	0.83
Local ablative therapy	10 (8%)	2 (6%)	8 (9%)	0.72
Radiation therapy	6 (5%)	1 (3%)	5 (6%)	0.67
Transcatheter arterial chemoembolization	9 (7%)	3 (8%)	6 (7%)	0.72

Patient characteristics are provided for the total cohort and separated for patients with low or high De Ritis ratio (>0.927), respectively. Both subgroups were compared using Fisher’s exact test or Wilcoxon rank-sum test.

**Table 2 cancers-13-00635-t002:** Univariable and multivariable Cox regression.

	Univariable Cox Regression			Multivariable Cox Regression		
Variable	Hazard Ratio	95% Confidence Interval	*p*-Value	Hazard Ratio	95% Confidence Interval	*p*-Value
Age (>66 years)	1.46	0.98–2.18	0.061	0.97	0.61–1.53	0.89
Sex (male)	0.82	0.55–1.22	0.32	-	-	-
ECOG score	-	-	0.074	-	-	0.22
0		*reference*			*reference*	
1	1.41	0.89–2.24	0.14	1.32	0.81–2.14	0.27
2	4.0	0.95–16.84	0.059	3.08	0.70–13.48	0.14
Charlson comorbidity index (CCI)	0.93	0.81–1.07	0.33	-	-	-
Primary tumor location	-	-	0.014	-	-	0.09
Gastrointestinal		*reference*			*reference*	
Pancreatic	1.8	1.11–2.94	0.018	1.51	0.92–2.47	0.11
Pulmonary	2.62	1.30–5.28	0.007	1.81	0.88–3.72	0.10
CUP	1.08	0.61–1.94	0.79	0.79	0.43–1.46	0.45
Metastatic spread						
Hepatic	0.59	0.26–1.35	0.21	-	-	-
Lymphonodal	1.33	0.74–2.40	0.34	-	-	-
Osseous	0.93	0.62–1.39	0.72	-	-	-
Peritoneal	1.17	0.73–1.87	0.51	-	-	-
Pulmonary	1.05	0.43–2.58	0.92	-	-	-
Functionality	0.83	0.55–1.26	0.39	-	-	-
Hedinger syndrome	1.35	0.55–3.33	0.52	-	-	-
De Ritis ratio (>0.927)	1.89	1.19–3.0	0.007	1.7	1.01–2.86	0.047
CgA (>204 μg/L)	2.16	1.35–3.46	0.001	2.05	1.24–3.39	0.005

## Data Availability

The data presented in this study are available on request from the corresponding author.

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
