# Peer review of "The Prognostic Value of the De Ritis Ratio for Progression-Free Survival in Patients with NET Undergoing [177Lu]Lu-DOTATOC-PRRT: A Retrospective Analysis"

_cancers, 2021, doi:10.3390/cancers13040635_

Round 1
Reviewer 1 Report
The topic of the article is very interesting considering the lack of literature data about De Ritis ratio in NETs, the simplicity and availability to obtain the parameter (AST/ALT ratio) and its prognostic value demonstrated also in other malignancies.
The analysis is clearly and in detail described, dealing with various endpoints in an exhaustive manner.
I have minor considerations for the authors:
- I would suggest moving the “Materials and Methods” paragraph immediately after the introduction, for a clearer reading of the results.
- In discussion section, in line 132-137, the results are reported again. Rather, I suggest to distribute them in the discussion, commenting them on the basis of literature data
- In Materials and Methods section, specify the radiopharmaceutical PET/CT used in the study and describe the scheduled doses of PRRT.
Reviewer 2 Report
The manuscript has evaluated the prognostic role of AST/ALT in NET treated with [177Lu]Lu-DOTATOC-PRR. The results and conclusion are well supported by data. The manuscript is comprehensive and well written. Please define all abbreviations at their first appearance (PFS).
detailed comments:
- Please provide the graph for the changes in the levels of AST, ALT, ALP, and CgA during the treatment.
- Radiation therapy may induce liver disease which typically presents 4–8 weeks after therapy and the changes in ALP are more pronounced that in AST and ALT, please provide the data for ALP for at least initial 10 weeks
- A reversible initial increase in AST and ALT has been reported with 177Lu-DOTATATE treatment, please discuss if the authors find the same or not.
- Similarly, please include the changes in LHD during the course of treatment
- Please provide the histological, imaging, CT scan or MRI imaging suggesting improvement with the therapy and no damage to liver and a decrease in liver and lung metastasis
- Was there any hematologic toxicity during the treatment
- Was there any radiogenic edema during the treatment
- An increase in size directly after PRRT has been reported, was there such findings during the study?
- Since this is a retrospective study and clinical investigations and imaging data is not available, please discuss these aspects in the discussion.
- Please expand all abbreviations on their first appearance.
